# Epigenetic weapons in plant-herbivore interactions: Sulforaphane disrupts histone deacetylases, gene expression, and larval development in *Spodoptera exigua* while the specialist feeder *Trichoplusia ni* is largely resistant to these effects

**Dana J. Somers, David B. Kushner, Alexandria R. McKinnis, Dzejlana Mehmedovic, Rachel S. Flame, Thomas M. Arnold***

Department of Biology, Program in Biochemistry and Molecular Biology, Dickinson College, Carlisle, PA United States of America

* arnoldt@dickinson.edu

## Abstract

Cruciferous plants produce sulforaphane (SFN), an inhibitor of nuclear histone deacetylases (HDACs). In humans and other mammals, the consumption of SFN alters enzyme activities, DNA-histone binding, and gene expression within minutes. However, the ability of SFN to act as an HDAC inhibitor in nature, disrupting the epigenetic machinery of insects feeding on these plants, has not been explored. Here, we demonstrate that SFN consumed in the diet inhibits the activity of HDAC enzymes and slows the development of the generalist grazer *Spodoptera exigua*, in a dose-dependent fashion. After consuming SFN for seven days, the activities of HDAC enzymes in *S. exigua* were reduced by 50%. Similarly, larval mass was reduced by 50% and pupation was delayed by 2–5 days, with no additional mortality. Similar results were obtained when SFN was applied topically to eggs. RNA-seq analyses confirm that SFN altered the expression of thousands of genes in *S. exigua*. Genes associated with energy conversion pathways were significantly downregulated while those encoding for ribosomal proteins were dramatically upregulated in response to the consumption of SFN. In contrast, the co-evolved specialist feeder *Trichoplusia ni* was not negatively impacted by SFN, whether it was consumed in their diet at natural concentrations or applied topically to eggs. The activities of HDAC enzymes were not inhibited and development was not disrupted. In fact, SFN exposure sometimes accelerated *T. ni* development. RNA-seq analyses revealed that the consumption of SFN alters gene expression in *T. ni* in similar ways, but to a lesser degree, compared to *S. exigua*. This apparent resistance of *T. ni* can be overwhelmed by unnaturally high levels of SFN or by exposure to more powerful pharmaceutical HDAC inhibitors. These results demonstrate that dietary SFN interferes with the epigenetic machinery of insects, supporting the hypothesis that plant-derived HDAC inhibitors serve as "epigenetic weapons" against herbivores.

**Data Availability Statement:** All raw sequence reads (fastq files) are available from the NIH GEO database (accession number GSE234351.)

**Funding:** This work was supported by grants from the National Science Foundation (award 2151434 to TA, DS, DK) and the Dickinson College Research and Development Committee (to TA). The funders had no role in study design, data collection and analysis, decision to publish, or preparation of the manuscript.

**Competing interests:** The authors have declared that no competing interests exist.

# Introduction

Plant chemical defenses act directly and indirectly against herbivores. Direct defenses include structural defenses such as waxes, thorns, hairs, and trichomes, as well as chemicals which act as toxins, deterrents, antifeedants, endocrine disrupters, photosensitizers, irritants and abrasives, immobilizers, allergens, and digestion reducers, among others [1–6]. Such direct defenses arose during the 350 million years plants have co-evolved with herbivores; they kill, injure, manipulate, or disrupt the development of herbivores [7]. Here we propose a new subcategory of plant defense, one capable of altering gene expression patterns of herbivores on a broad scale by sabotaging their epigenetic control systems. We focus on plant products that inhibit the activity of nuclear histone deacetylase (HDAC) enzymes in herbivores, disrupting the balance of acetylation and deacetylation of nuclear histone proteins in the animal's tissues. These substances alter DNA-histone binding properties, creating new regions of loosely packed euchromatin and tightly packed heterochromatin, thereby altering the accessibility of large numbers of genes [8–11]. HDAC inhibitors are common in plants [12–14] and their effects on humans and other mammals have been investigated in clinical trials [13,15–22]. However, little is known about the roles of these botanical HDAC inhibitors in nature. We suggest that they act as epigenetic weapons, disrupting the development, phenotype, behavior, dispersal, or reproduction of herbivorous insects via epigenetic mechanisms. To test this idea, we examined the effects of sulforaphane (SFN), an HDAC inhibitor produced by cruciferous plants, on two common lepidopteran grazers which are important agricultural pests.

## 1. The importance of epigenetic systems in insects

Epigenetic systems are ubiquitous in metazoans where they alter gene expression *via* DNA methylation, regulatory action of non-coding RNA, or modification of histone proteins [23–25]. These chromatin modifications open and close regions of the genome and alter the ability of transcription factors to access genes. As a result, they can orchestrate large-scale changes in gene expression, generating new metastable phenotypes in individuals [26–28]. For example, epigenetic systems can direct normal development and aging, coordinate stress responses, and facilitate rapid adaptation to changing environments. Altered phenotypes may persist long after a stimulus has passed [8,13,24–27,29]. New phenotypes generated by these epigenetic systems are termed epigenotypes [30]. In insects, where phenotypic changes can be especially dramatic, they are often referred to as polyphenisms. Acquired epigenetic states can sometimes be inherited, altering the phenotype of offspring in ways that may (or may not) improve their fitness [24,31,32]. In short, epigenetic mechanisms are an important source of phenotypic plasticity for individuals. They may also confer benefits at the level of populations by facilitating adaptation to changing conditions when genetic variation is lacking, such as in small, bottlenecked, or asexual populations [24,33].

Relatively little is known about the epigenetic machinery of insects [34–37]. However, recent studies demonstrate that they can be important in development, sex determination, morphology, behavior, life history cycles, longevity, pathogen resistance, and immune priming as well as eusocial caste structure [34,35,38–40]. For instance, various epigenetic systems control development, physiology, and behavior in *Drosophila* [41]; determine the sex of silk moths (*Bombyx*) [42]; control the size and shape of insect armament in beetles [43]; alter juvenile hormone levels [39]; and may alter the duration and direction of flight in the cotton bollworm, *Helicoverpa armigera* [36]. Epigenetic mechanisms determine the caste of individuals in eusocial insects, including bees and ants [35,44–46]. Clearly, epigenetic systems can initiate phenotypic plasticity in response to changing conditions or, in some cases, exposure to specific chemicals. There are some examples of epigenetic effects spanning generations in insects,

though these reports are relatively rare. For example, *Manduca sexta* epigenetic mechanisms facilitate trans-generational immune priming against pathogenic bacteria [40].

Three epigenetic systems have been commonly examined in insects [24,34,35,37]: DNA methylation, non-coding RNA, and histone modifications.

**DNA methylation.**   In most animals methylation of DNA tends to repress transcription and the majority of CpG dinucleotide sites are methylated at any given time [36]. However, DNA methylation may not be as common, or as important, in insects. For example, DNA methylation was not observed in *Drosophila* and is thought to be sparse or absent in many other insect taxa, leading some to conclude that DNA methylation has at most a minor role in insects [35,36,38,47,48]. Some insects possess the enzymes required for DNA methylation but others, including *Drosophila* and some lepidopterans, do not [35,36]. There are a few reported examples of DNA methylation impacting insect fitness [36,38] but the importance of DNA methylation in insects is a subject of debate [34].

**Non-coding RNA.**   Non-coding RNAs may bind to DNA and alter chromatin to yield epigenetic effects and some have roles in insects [35,38]. For example, they may be protective against mutations caused by transposable elements; they occasionally determine insect sex, as in *Bombyx*; they alter DNA binding to histones; and they may be maternally transmitted to offspring [35]. More work is needed to understand the functions of non-coding RNAs in insects.

**Histone modifications.**   Gene expression may be controlled by altering DNA binding to histone proteins [9,11]. Histone acetyltransferases modify histone subunits by adding acetyl groups, creating regions of loosely packed euchromatin that makes genes more accessible for transcription. Conversely, HDACs remove acetyl groups, leading to tightly packed heterochromatin that is less accessible for transcription [8–11]. Here we focus on plant products that inhibit HDACs, for several reasons. First, plants produce an assortment of natural products known to inhibit HDAC enzymes in mammals [13,15,49–53]. Examples include allicin from garlic; curcumin from turmeric; apigenin, genistein, and quercetin from fruits; sinapinic acid from mustard seeds; resveratrol from grapes and wine; SFN from cruciferous vegetables; caffeic acid and catechins from foliage and tea; protocatechuic aldehyde from wine stopper cork; diallyl disulfide from garlic; zerumbone from ginger; ursolic acid from basil; and butein and kaempferol from a variety of plants [12–14]. This evidence comes primarily from medical studies focusing on mice and cultured cells as well as from clinical trials. Second, there is evidence that HDAC inhibitors from other, non-plant sources can affect insects quite dramatically. For example, phenyl butyrate in royal jelly is a natural HDAC inhibitor and determines caste and behavior in honey bees and ants [35,38]. The related HDAC inhibitor, sodium butyrate, improves learning in honey bees [54]. In cultured cells of *Spodoptera frugiperda*, sodium butyrate causes mitochondrial dysfunction, oxidative stress, and cell death, [55] while also altering susceptibility to viruses [56]. Exposure to trichostatin A (TSA), a pharmaceutical HDAC inhibitor, alters the size and shape of mandibles in the broad-horned flour beetle, *Gnatocerus cornutus* [43]. It also alters histone acetylation in the brains of *Camponotus floridanus* ants, resulting in changes to their behavior [44]. RNAi silencing of HDAC genes alters histone acetylation in the brown planthopper, *Nilaparvata lugen*¸ affecting the development of ovaries and ovipositors in females and preventing males from making courtship songs [57].

There is an obvious disconnect between these two sets of studies. One set highlights the many HDAC inhibitors produced by plants and examines their use in medicine. The other set shows that HDAC enzymes can affect insects but focuses on HDAC inhibitors from non-plant sources. We are not aware of any study connecting these two sets of observations, *i.e.*, testing the potential impact of botanical HDAC inhibitors on the development, physiology, or behavior of herbivores. As a result, little is known about the roles of botanical HDAC inhibitors in

nature. This gap in knowledge is surprising since many herbivorous insects consume large amounts of HDAC inhibitors when feeding upon their host plants.

## 2. HDAC inhibitors

There are 18 recognized HDAC enzymes, organized into four classes [9,12,35,58,59]. The "classic" HDACs are zinc-dependent hydrolases. These include class I enzymes (HDACs 1–3, 8), which are ubiquitous and essential for cell proliferation and survival; class IIa enzymes (HDACs 4, 5, 7, 9) have weak activity and move in and out of the nucleus; class IIb enzymes (HDACs 6, 10) act upon non-histone proteins in the cytosol; class IV enzymes (HDAC 11) have poorly-defined functions. Other HDAC enzymes include the sirtuins, class III NAD$^+$ dependent enzymes (SIRT 1–7) that are involved in aging, transcription, apoptosis, inflammation, axonal degeneration, stress resistance, metabolic regulation, and energy production.

HDAC inhibitors impede the activity of one or more classes of HDAC enzymes [11]. Many are common in nature; some are FDA-approved therapeutics used in oncology and neurology [59]. Pharmaceutical HDAC inhibitors include TSA, romidepsin (Istodax), and suberanilohydroxamic acid (Vorinostat). Less powerful HDAC inhibitors include certain ketones, sodium- and phenyl-butyrates, and the anti-seizure medication valproic acid [13]. Plants produce a variety of HDAC inhibitors, such as SFN, an isothiocyanate from cruciferous vegetables, which inhibits class II HDAC enzymes in mammals [13,49,53]. SFN has health benefits in humans [51,60–63]. For example, it initiates anti-oxidant and anti-inflammatory responses and restores proteasome function [64,65]. In cancer cells, it can trigger cell cycle arrest and apoptotic cell death thereby inhibiting tumor growth [66,67]. SFN can cross the blood-brain barrier and has neuroprotective effects [64,68,69]. In mammals SFN is easily absorbed and distributed in the body: after consumption of broccoli, for example, the concentration of SFN in human plasma increases quickly and plateaus within several hours [69–72]. The potential roles of SFN in nature, however, have not received much attention.

## 3. Testing HDAC inhibitors as potential plant defenses in a model system

SFN is produced by plants in the order Brassicales (the cruciferous or mustard oil plants) including cabbages, broccoli, Brussels sprouts, bok choy, cauliflower, rapeseed, watercress, and *Arabidopsis* [6,62,73–76]. This group includes the most important cultivated leafy vegetables [67]. When these plants are chewed or crushed, a "mustard oil bomb" reaction is generated in which glucosinolates are hydrolyzed by plant myrosinases (ß-thioglucoside glucohydrolases) to generate nitriles, isothiocyanates, thiocyanates, oxazolidine-2-thiones, and epithionitriles [6,73,74,77,78]. These products can serve as defenses against herbivores [67,79–88] and influence the taste and flavor of cruciferous plants. SFN is the hydrolysis product of one particular glucosinolate, glucoraphanin (4-(methylsulfinyl) butyl glucosinolate) [61,62,64]. Most insect herbivores feeding on cruciferous plants would be exposed to SFN with every bite, though some possess adaptations to deactivate or redirect the mustard oil bomb reactions [80,88–90]. These substances are known to have a variety of deterrent effects against a range of grazers; however, as far as we are aware, the impacts of SFN on the epigenetic systems of insect herbivores have not been investigated.

Here we investigated the impact of SFN on two lepidopteran species, one generalist (the beet army worm, *Spodoptera exigua*) and one specialist (the cabbage looper, *Trichoplusia ni*), both of which are agricultural pests and consume cruciferous plants in nature. *S. exigua* has a broad host range; in contrast, *T. ni* grazes most often on members of the Brassicales [91], including those that generate SFN [6,79,87,92]. Our experiments tested the hypothesis that SFN interferes with the epigenetic machinery of herbivorous insects.

## Methods

### Experimental approach

We conducted a series of experiments to determine if dietary SFN (a) affects development, (b) inhibits the activity of nuclear HDAC enzymes, and/or (c) alters patterns of gene expression in the beet armyworm (*S. exigua*) and the cabbage looper (*T. ni*). For each experiment, larvae were allowed to feed *ad lib* on an artificial diet (AD) containing natural concentrations of SFN. Others consumed pharmaceutical HDAC inhibitors, such as TSA and romidepsin (ROM), which served as response controls (Fig 1). We monitored the development of larvae and in some cases adult moths. Differential development was analyzed using Student's t-tests or ANOVAs with Holm-Sidak posthoc tests. Non-parametric alternatives, including Mann-Whitney U tests and ANOVAs on ranks with Dunn's pairwise comparisons, were used when datasets were not normally distributed. Tissues were also collected to quantify HDAC enzyme activities and to identify gene expression patterns *via* RNA-seq.

### Insects and treatments

*T. ni* and *S. exigua* larvae were obtained from Benzon Research (Carlisle, Pennsylvania, USA). First instar larvae were raised individually in 5 ml wells on an AD appropriate for each species with antibiotics at 25°C and 50% humidity, unless otherwise indicated.

The HDAC inhibitors SFN, TSA, and ROM were obtained commercially from Millipore Sigma (Burlington, Massachusetts, USA) and BioVision Incorporated (Waltham, Massachusetts, USA). Each HDAC inhibitor was solubilized in ethanol and added to each well to cover the surface of the AD. The ethanol was fully evaporated for 8h at 20°C. One first instar larva was transferred into each well on Day 0. Larvae fed *ad lib* and generally consumed half of the AD by Day 7. We employed a variety of negative controls, including larvae feeding on AD alone and larvae feeding on AD treated with the ethanol carrier solvent only. In a few cases we raised a second generation by allowing larvae to pupate, emerge as moths, and mate within treatment groups. In these cases, second-generation larvae were reared from the eggs on AD without HDAC inhibitors.

The amount of SFN utilized in these experiments is similar to what is ingested during natural feeding. For example, we typically used SFN concentrations of 0–70 mM, applied as 30–40 μl aliquots to each well containing AD. This range of concentrations is similar to that of cruciferous plant foliage and lower than levels reported in buds, flowers, sprouts, or seeds. The doses are also comparable to those used in medical studies [93].

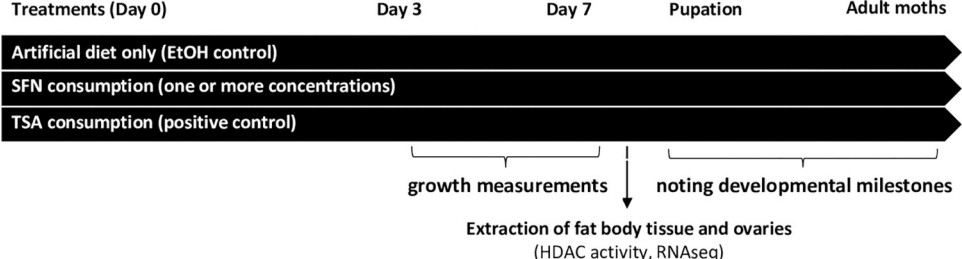

**Fig 1. Basic experimental design.** For each species, first instar larvae in individual wells consumed various concentrations of SFN in an AD. Larvae in negative control groups consumed AD treated only with the carrier solvent (ethanol), which was evaporated prior to feeding. Larvae in positive control groups consumed AD supplemented with a pharmaceutical HDAC inhibitor, usually TSA. Larvae were raised at 25°C and 50% humidity, unless otherwise noted. Biometric data, such as mass, length, and width, were recorded for individual larvae. Tissues were isolated from a subset of larvae for HDAC enzyme activity assays and gene expression analyses by RNA-seq. Remaining larvae were allowed to develop normally. Adjustments made for some specific experiments are described in the methods section.

We considered the possibility that SFN and TSA could be distasteful, suppressing feeding and, therefore growth rates. We did not observe this to be the case in any experiments. Nonetheless, it seemed prudent to run parallel experiments wherein these substances were applied topically, rather than *via* the diet. As a result, we also conducted similar experiments wherein HDAC inhibitors were applied topically to eggs, using DMSO as a penetrating solvent.

## Sample collection

Larvae developed in individual wells until pupation, approximately Day 14 depending on species, treatment, and temperature. Larval masses, larval widths, and survivorship were recorded. Fat body tissues, and in some cases ovaries, were collected by microdissection from a subset of larvae, approximately four days before control larvae would start pupation. We focused on these tissues because they are important in development and reproduction, because they can be obtained cleanly from larvae that remain smaller than usual, and because SFN is expected to reach them relatively quickly. We did not examine midgut tissue, commonly examined in similar studies, because gut tissue would likely be contaminated with HDAC inhibitors from the artificial diet, making accurate measurements of HDAC enzyme activity problematic. Tissues used to quantify HDAC enzyme activities were stored immediately at -80°C [94]. Tissues for RNA-seq analyses were preserved in RNA*later* (Invitrogen, San Diego, California, USA; ThermoFisher Scientific, Waltham, Massachusetts, USA) using 1 ml per 100 mg tissue, and stored at -80°C.

## HDAC enzyme activity

Extracts of nuclear and cytoplasmic proteins were made from isolated tissues (BioVision K266-25, Waltham, Massachusetts, USA). HDAC enzyme activity was determined using an HDAC Activity Fluorometric Assay Kit (BioVision K330-100, Waltham, Massachusetts, USA) and normalized to protein concentration in each extract.

## RNA extraction, library construction, and sequencing

Larvae from each experiment group were dissected to isolate fat body tissue. Fat bodies from three to five individual larvae were pooled per replicate. Thawed samples were centrifuged at 10,000 x *g* for 20 seconds and RNA*later* was removed from pelleted tissue. Total RNA was extracted from the tissue twice using Buffer A (50 mM sodium acetate pH 5.2, 10 mM EDTA, 1% SDS) saturated phenol heated to 65°C, followed by phenol/chloroform (1:1) extraction. Extracts were ethanol precipitated, resuspended in ddH$_2$O and ethanol precipitated again. Final pellets were resuspended in ddH$_2$O to a concentration of 500–1500 ng/ml. RNA concentration was determined using a DS-11 spectrophotometer (DeNovix, Wilmington, Delaware, USA). Total RNA quality was assessed using RNA Nano Chips and a 2100 BioAnalyzer (Agilent, Santa Clara, California, USA) and Qubit RNA High Sensitivity Assay (ThermoFisher, Waltham, Massachusetts, USA), followed by High Sensitivity RNA ScreenTape Analysis (Agilent, Santa Clara, California, USA).

Library construction and sequencing was performed by Admera Health Biopharma Services (South Plainfield, New Jersey, USA). Poly(A) selection was performed using the NEBNext Poly(A) mRNA Magnetic Isolation Module (New England Biolabs, Ipswich, Massachusetts, USA) following the manufacturer's protocols. RNA-seq libraries were generated using the NEBNext Ultra II RNA Library Prep Kit for Illumina (New England Biolabs, Ipswich, Massachusetts, USA) following manufacturer's protocols. Libraries were quality checked using the Qubit dsDNA High Sensitivity Assay (ThermoFisher, Waltham, Massachusetts, USA), followed by High Sensitivity DNA ScreenTape analysis (Agilent, Santa Clara,

California, USA) and qPCR (KAPA SYBR FAST qPCR Master Mix (2X) Kit; Roche, Basel, Switzerland). Libraries were standardized to equal molar ratios and then pooled. Paired-end 150-bp reads were generated using the NovaSeq S4 platform (Illumina, San Diego, California, USA). All raw sequence reads were deposited in the NIH GEO under project number GSE234351.

### RNA-seq processing and analysis

Reads for all RNA-seq experiments were quality checked with FastQC (v0.11.9) [95] and were processed with Trimmomatic (v0.39) [96] using the following parameters: LEADING:5, TRAILING:5, MAXINFO:36:0.2, MINLEN:50. *T. ni* reads were aligned to the *T. ni* Cornell-1 isolate genome (obtained from the *T. ni* Genome Database on 2022-05-25) and *S. exigua* reads were aligned to the *S. exigua* TB_SE_WUR_2020 isolate genome (WGS JACEFF01 obtained from the NCBI Genome Database on 2022-06-06) [97,98]. Reads were aligned using HISAT2 (v2.2.1) with the default parameters, and alignments were quality checked using QualiMap (v2.2.1) [99]. featureCounts implemented in the Rsubread package (v2.12.2) was used to calculate read counts for each gene using the appropriate species-specific annotations [100]. Low-count genes (mean counts-per-million [CPM] across all samples > 10) were filtered and read counts were normalized using the trimmed mean of M-value normalization method. Differential expression analysis was performed with edgeR (v3.40.2) [101,102] using a general linearized model comparing the mean expression for all pairwise combinations of treatments, pairing triplicate samples, and taking the Benjamini and Hochberg false discovery rate (FDR) < 0.05 as significant [103].

One-to-one orthologs between *T. ni* and *S. exigua* were assigned using the reciprocal best hit method with BLAST [104]. Hierarchical clustering was performed using Cluster 3.0 [105] and visualized in JavaTreeView [106]. KEGG and GO annotation terms were assigned to each genome using eggNOG (v2.1.7) [107]. Functional enrichment analysis of orthologous gene lists was performed with clusterProfiler (v4.6.0) [108], using a hypergeometric test, taking as background the set of genes in each species that were retained after filtering for low count genes, and taking the Benjamini and Hochberg FDR < 0.05 as significant [103].

## Results

### The growth of the generalist feeder, *S. exigua*, but not the specialist feeder *T. ni*, was negatively affected by HDAC inhibitors

SFN slowed the development of *S. exigua* larvae in a dose-dependent manner. *S. exigua* larvae feeding upon the highest doses of SFN were, on average, less than half the size of those in the control groups at Day 11 (Fig 2A). The treated larvae seemed otherwise healthy: they readily fed on diets containing SFN and pupated normally and there was no significant mortality. In contrast, the development of *T. ni* larvae was generally unaffected by SFN. For example, SFN had no impact on *T. ni* larval mass at Day 7 (P = 0.502) or Day 11 (P = 0.780; Fig 2A) and there were no obvious changes in feeding, behavior, or pupation.

### Topical application of SFN elicited responses similar to feeding

We observed similar results when SFN was applied topically to egg masses (Fig 2B). Direct SFN treatment inhibited the development of *S. exigua* (Student's t-test; P<0.001). In contrast, we observed that *T. ni* larvae from SFN-treated eggs grew *faster*, compared to controls (Student's t-test; P<0.001).

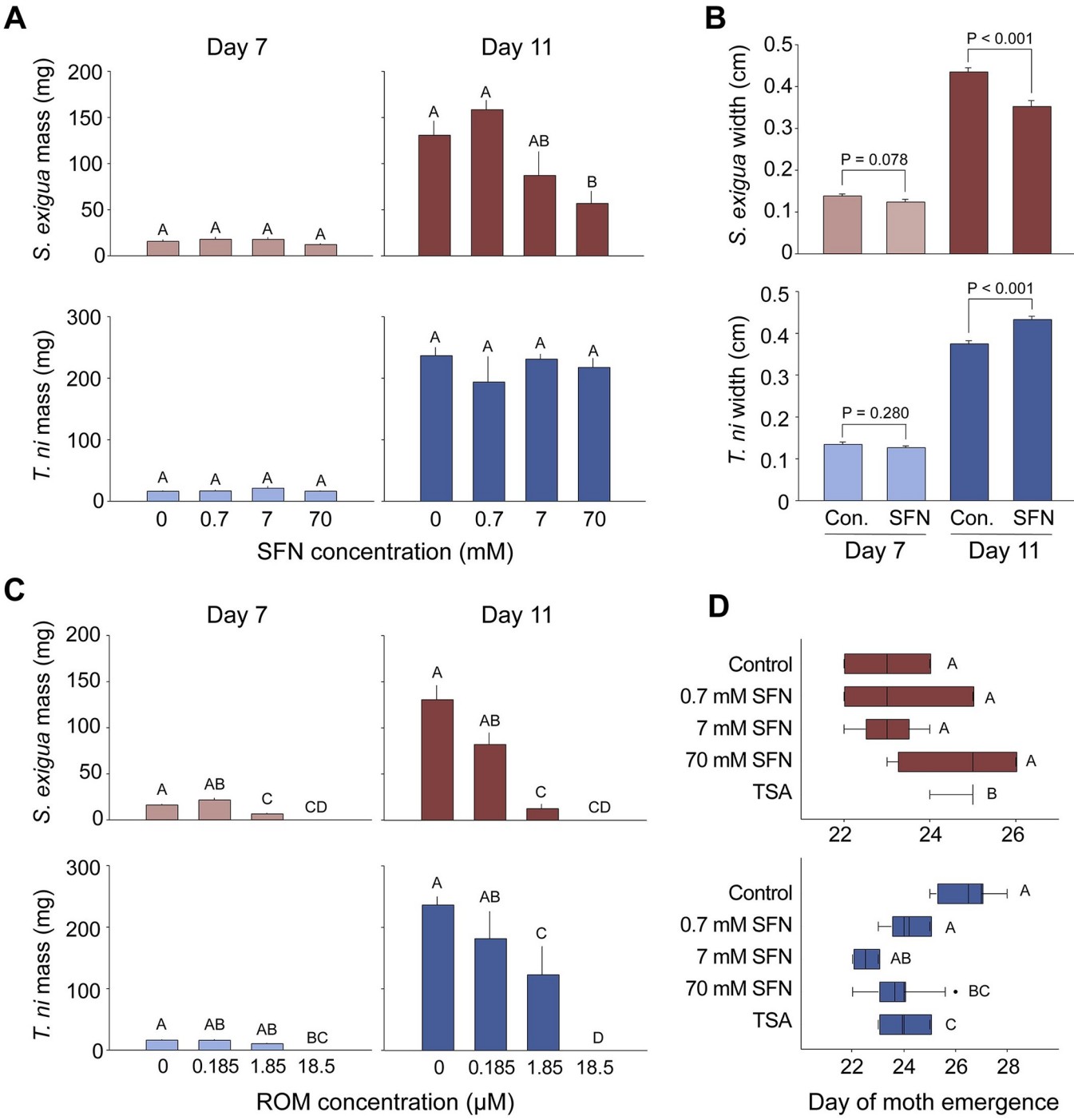

**Fig 2. Effects of HDAC inhibitors on development of *S. exigua* and *T. ni* larvae.** (A) The mass of *S. exigua* larvae was reduced by consumption of SFN at Day 11 (One-way ANOVA, P = 0.006; red bars). No significant effect on larval mass was observed for *T. ni* (One-way ANOVA, P = 0.780; blue bars). (B) By Day 11, the width of *S. exigua* larvae was reduced when eggs were directly exposed to SFN (Student's t-test, P<0.001; red bars). In contrast, the width of *T. ni* larvae was not reduced but instead increased by Day 11 (Student's t-test, P<0.001; blue bars). Con. indicates use of the DMSO carrier solvent. (C) *S. exigua* larval mass was decreased at both Day 7 (One-way ANOVA, P<0.001) and Day 11 (One-way ANOVA, P<0.001) after exposure to the powerful HDAC inhibitor ROM. *T. ni* larval mass was also decreased at Day 7 (One-way ANOVA, P<0.001) and Day 11 (One-way ANOVA, P<0.001). (D) The emergence of *S. exigua* moths was not delayed by the consumption of SFN, although TSA delayed moth emergence by about two days (One-way ANOVA, P = 0.003). The emergence of *T. ni* moths was not delayed by SFN or TSA. Instead, both TSA and SFN accelerated moth emergence significantly (One-way ANOVA, P<0.003). Control indicates the use of the EtOH carrier solvent. All data represent the mean +/- SE; letters indicate the results of post hoc tests; bars with the same letters are not significantly different at α = 0.05 according to pairwise multiple comparisons.

## Stronger HDAC inhibitors challenge the resistance of *T. ni*

*T. ni* appeared to be resistant to the effects of SFN at natural concentrations, so we attempted to overwhelm this resistance by exposing larvae to ROM, one of the strongest pharmaceutical HDAC inhibitors. We observed that the growth of *T. ni* larvae was reduced dramatically by ROM in a dose-dependent manner (Fig 2C). ROM treatments also delayed pupation up to 5 days and delayed the emergence of adult moths up to 10 days, on average. Only the highest treatment of 18.5 μM ROM caused significant mortality. Nevertheless, consistent with the aforementioned results, *T. ni* was less impacted than *S. exigua*.

## Long-term effects

We observed some impacts of HDAC inhibitors on later developmental milestones. In the SFN treatment groups, the emergence of *S. exigua* moths was often delayed by a few days. Stronger HDAC inhibitor drugs slowed the emergence of *S. exigua* moths more dramatically (Fig 2D). Interestingly, the emergence of *T. ni* moths was not delayed. Instead, it was *accelerated* by approximately two days in both TSA and the highest concentrations of SFN (Fig 2D). Despite these effects on larval development, we never detected differences in pupal mass or the size, appearance, and behavior of adult moths for any treatment group.

## SFN inhibits HDAC enzymes in *S. exigua* but not *T. ni*

We expected the consumption of SFN and TSA to inhibit the activity of HDAC enzymes in larval tissues, as they do in mammals, and indeed this was the case for *S. exigua*. In this species, HDAC enzyme activity was reduced by ~50% by SFN (Fig 3A). In contrast, HDAC enzyme activities in *T. ni* tissues were not inhibited by exposure to SFN.

We wondered if the HDAC enzymes of *T. ni* were naturally resistant to SFN. To test this, we extracted nuclear and cytoplasmic HDAC enzymes from both species and exposed them to SFN *in vitro*. The enzymes extracted from both species were inhibited equally by SFN (Fig 3B) suggesting that HDAC enzymes from *T. ni* were not inherently more resistant to SFN.

## The effects of SFN on *S. exigua* are most apparent at higher temperatures

Changes in temperature impacted larvae development, as expected (Fig 4). For both species, larval growth rates increased markedly as temperatures increased from 23 to 29˚C. Consistent with previously described experiments where larvae consumed SFN, development of *S. exigua* was reduced by consumption of SFN; this effect was most apparent at higher temperatures. For example, at 23˚C the mass of *S. exigua* larvae was reduced 20% from 65 mg to 52 mg, on average, whereas at 29˚C mass was reduced 30% from 226 mg to 157 mg, on average. In these experiments, consumption of SFN failed to affect the development of *T. ni* at any temperature.

## SFN induces large-scale changes in gene expression

To investigate the consequences of HDAC inhibition in larvae consuming SFN or TSA (S1 Fig), we profiled transcriptome changes in *S. exigua* and *T. ni* fat body tissues, in biological triplicate. We applied a linear model to identify genes differentially expressed in each species, in response to each HDAC inhibitor. In all, we identified 1,792 and 2,454 genes whose expression was significantly altered in *S. exigua* and *T. ni*, respectively, after consumption of either SFN or TSA (S2 Fig). To directly compare the response between species, we identified one-to-one orthologs. In both species, the expression patterns were similar: approximately 85% of significantly differentially expressed genes are repressed in response to either SFN or TSA, while approximately 15% genes are induced (Fig 5B and 5C). The genes repressed in response to

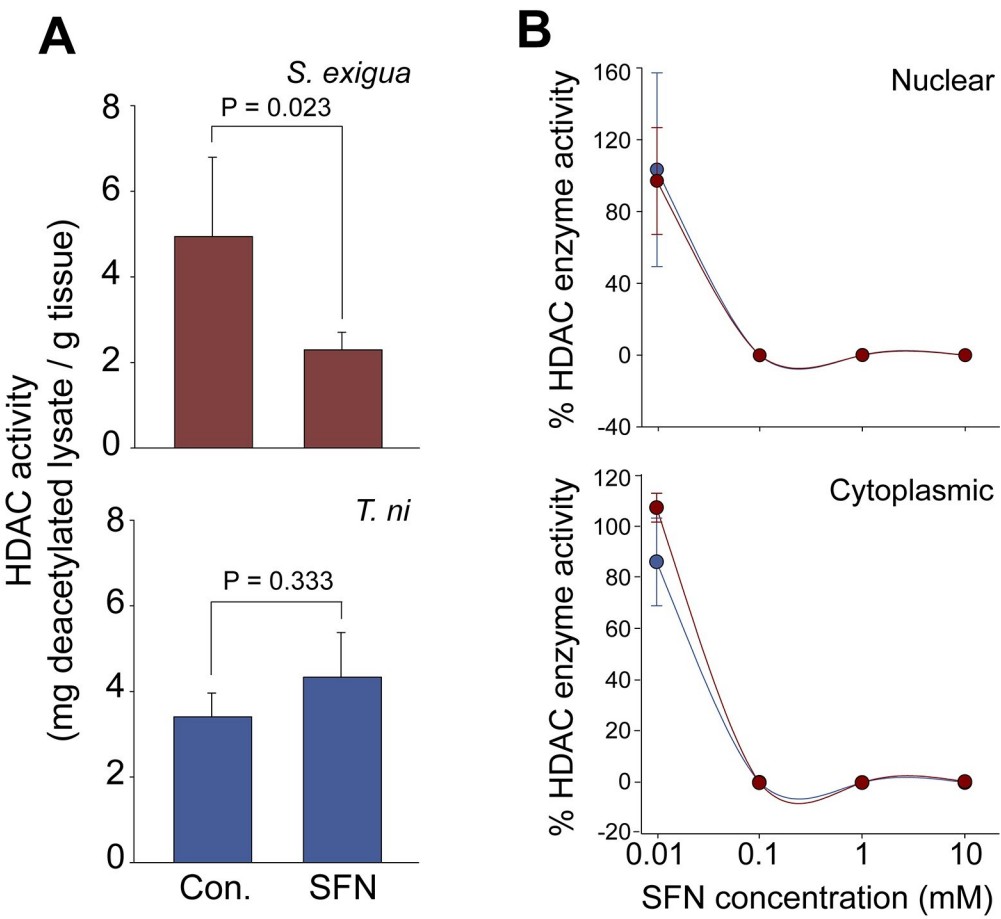

**Fig 3. Effect of SFN on HDAC enzymes in *S. exigua* and *T. ni*.** (A) The activity of HDAC enzymes in nuclear extracts was reduced in *S. exigua* after consumption of SFN (Student's t-test, P = 0.023; red bars). In contrast, enzyme activity was not reduced in *T. ni* (Student's t-test, P = 0.333; blue bars). (B) Nuclear and cytoplasmic HDAC enzymes were extracted from *S. exigua* (red) and *T. ni* (blue), and their activities were quantified *in vitro* in the presence of various concentrations of SFN. Con. indicates use of the EtOH carrier solvent in (A) while the leftmost data points represent the control in (B). Data represent the mean of 3–5 replicates +/- SE.

SFN strongly overlap the genes repressed in response to TSA. However, we also identified 181 and 409 genes in *S. exigua* and *T. ni*, respectively that showed species-specific repression. The genes induced in response to SFN and TSA have less overlap between species. Importantly, although the response patterns were similar for both species, the *magnitude* of the response to these HDAC inhibitors was greater in *S. exigua* compared to *T. ni* (Fig 5A).

To explore the potential physiological effects of differential gene expression elicited by SFN and TSA, we clustered the orthologous expression responses and identified enrichment of KEGG pathways in specific gene clusters (Table 1). Both species exhibited reduced expression of genes associated with energy conversion pathways, including pyruvate metabolism, carbon metabolism, oxidative phosphorylation, the TCA cycle, glycolysis, and gluconeogenesis. Again, the degree to which these genes were repressed was less in *T. ni* as compared to *S. exigua*. In addition, both species showed increased expression of genes associated with ribosome biosynthesis and the production of cuticular proteins. Finally, a small cluster of 146 genes that were induced in *T. ni* but repressed in *S. exigua* showed a significant enrichment for genes involved in fatty acid degradation.

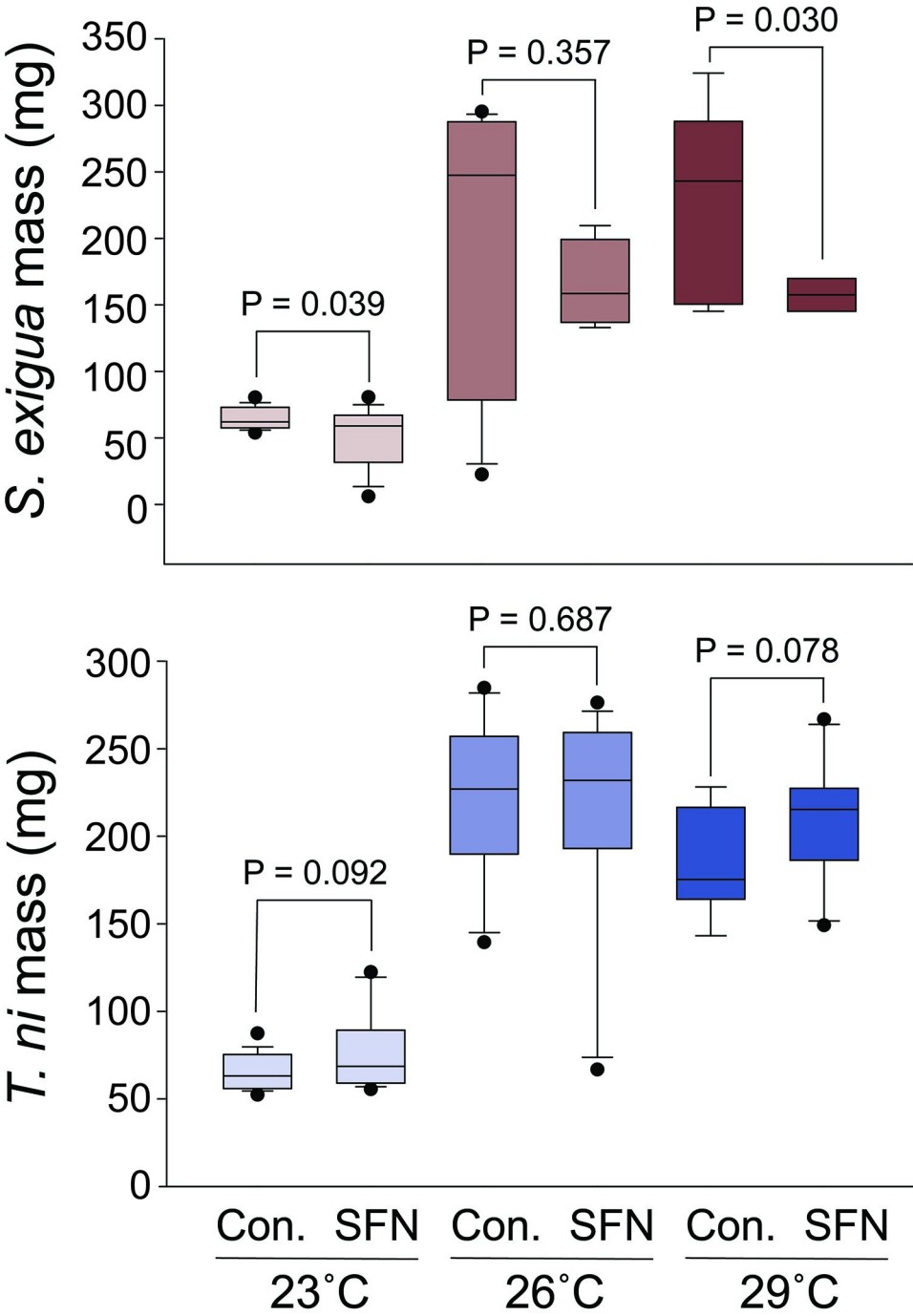

**Fig 4. Effect of temperature on larvae consuming SFN.** Exposure to SFN affected the mass of *S. exigua* larvae at various temperatures. The impact of SFN on *S. exigua* was most apparent at higher temperatures. For *T. ni*, larval mass was not reduced by SFN at any temperature. Con. indicates use of the EtOH carrier solvent. P values are the results of Student's t-tests.

It is possible that HDAC inhibitors such as SFN and TSA not only inhibit HDAC enzyme activity directly but also repress the expression of genes encoding for these enzymes, a type of "double inhibition". To investigate this, we specifically examined the expression of known

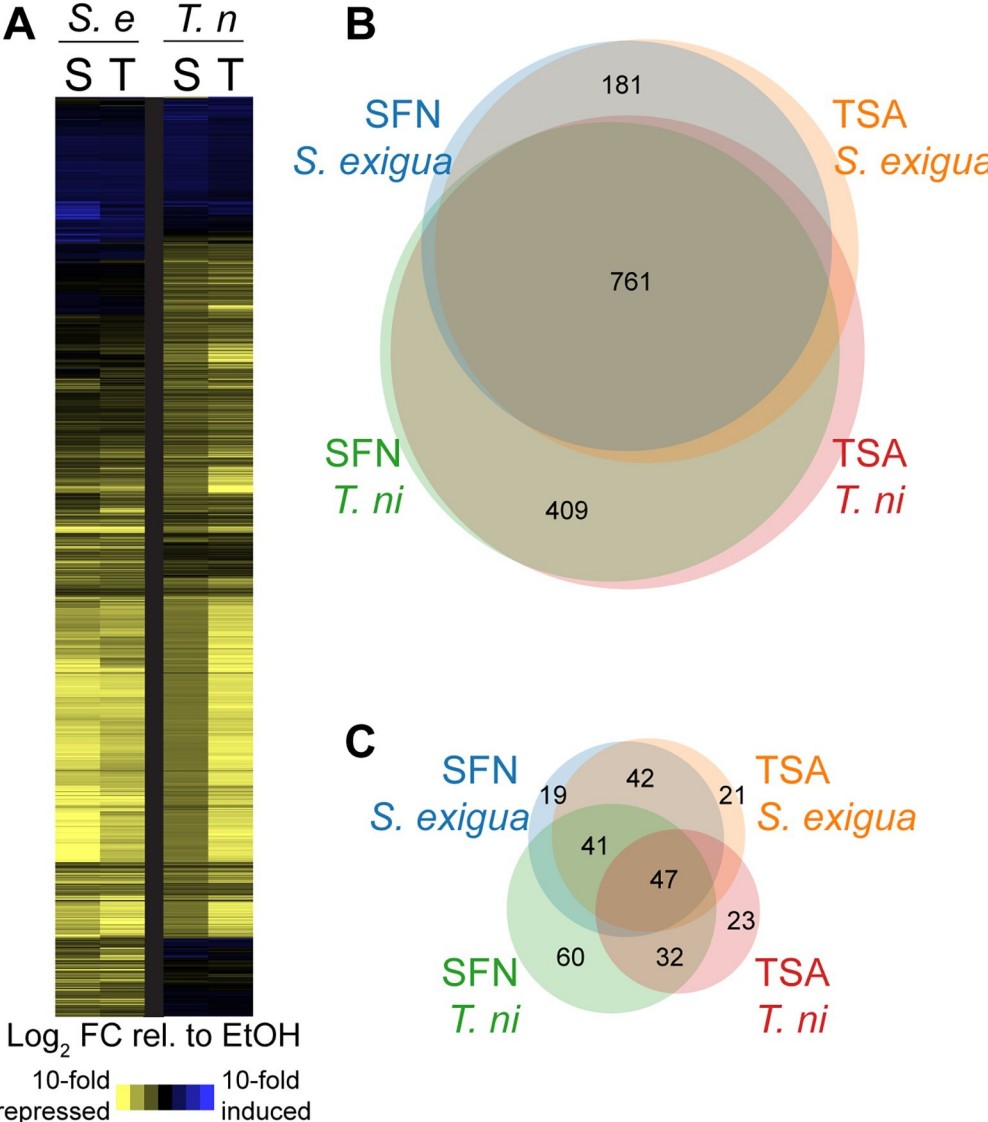

**Fig 5. Consumption of SFN altered the expression of genes in fat body tissues of *S. exigua* and, to a lesser degree, *T. ni*.** (**A**) Hierarchical clustering based on Euclidean distance of 1,721 orthologous genes significantly differentially expressed after exposure to either SFN (S) or TSA (T) in *S. exigua* (*S.e*) and *T. ni* (*T.n*). Values represent the log$_2$ fold-change relative to an EtOH control. Venn diagrams represent the overlap of orthologous genes significantly repressed in response to SFN or TSA (n = 1,615; **B**) or significantly induced in response to SFN or TSA (n = 307; **C**). For overlap categories representing less than 5% of the total number of genes, the number is not denoted in the Venn diagram.

HDAC genes in these species of insects. We identified six HDAC genes in each of our species: HDAC2 & 3 (Class I), HDAC6 & 7 (Class II), and HDAC11 (Class IV). Only HDAC11 was significantly repressed in our experiment, in both *S. exigua* and *T. ni* (S1 Table).

## Iflaviruses

Our RNA-seq analyses confirmed the presence of iflavirus 1, a positive-strand RNA virus, in the fat body tissues of *S. exigua*. This result was not surprising. Iflaviruses are common in lab-cultured populations [109] and field populations [110,111] of insects. In many insect species, including *S. exigua*, iflaviruses produce "covert infections" that are not lethal and do not

**Table 1. Enrichment for KEGG pathways in clusters of genes with different expression patterns.**

| Function | P value [a] | Number of Genes in Cluster | Number of Genes in Genome | Fold Enrichment over Background |
|---|---|---|---|---|
| **Repressed in both species** | | | | |
| Pyruvate metabolism | 5.00e-09 | 19 | 51 | 5.94 |
| Oxidative phosphorylation | 1.14e-05 | 27 | 148 | 2.91 |
| Carbon metabolism | 3.90e-05 | 25 | 141 | 2.83 |
| Citrate cycle (TCA cycle) | 7.72e-05 | 12 | 40 | 4.79 |
| Biosynthesis of amino acids | 0.0012 | 16 | 86 | 2.97 |
| Cysteine and methionine metabolism | 0.0057 | 10 | 45 | 3.54 |
| Glycolysis / Gluconeogenesis | 0.0191 | 10 | 53 | 3.01 |
| Glycerophospholipid metabolism | 0.0450 | 11 | 70 | 2.51 |
| **Induced both species** | | | | |
| Ribosome biogenesis in eukaryotes | 6.41E-35 | 31 | 71 | 21.15 |
| **Repressed in *S. exigua* and induced in *T. ni*** | | | | |
| Fatty acid degradation | 0.0306 | 4 | 42 | 9.95 |

[a] P value from Bonferroni-corrected hypergeometric test.

produce obvious signs of disease [109,112]. Carballo *et al.* noted that they are often detected serendipitously during transcriptomic studies [113], as was the case in our experiments. However, a few recent studies have suggested that there may be direct or indirect effects of iflavirus infection, for example interactions with other viruses or predators known to affect insect fitness [113]. We wondered if the consumption of HDACi in our experiments affected the abundance of iflavirus RNA. It did not; our analyses indicated that viral RNA levels were not impacted by the consumption of HDAC inhibitors by its host.

## Discussion

### 1. Sulforaphane can act as an epigenetic weapon against *S. exigua*

SFN, a natural HDAC inhibitor produced by cruciferous plants, disrupted the development of the beet armyworm, *S. exigua*, when consumed at natural concentrations. In general, SFN slowed larval growth by ~50%. Interestingly, there was no significant mortality. Larvae fed normally, were otherwise healthy, and pupated to produce adult moths. In this species, consumption of SFN inhibited the activity of HDAC enzymes in the nucleus and the cytoplasm of fat body tissues by ~50% and resulted in large scale changes in patterns of gene expression. We detected 1,792 differentially expressed genes in *S. exigua* larvae consuming SFN. In general, genes associated with central energy production pathways were downregulated while genes associated with ribosome construction and cuticle formation were upregulated in fat body tissues. The method of SFN treatment did not seem to matter: similar responses were observed when SFN was consumed in the diet and when eggs were treated topically with SFN, supporting the hypothesis that SFN acts directly as a HDAC inhibitor rather than affecting feeding rates. The effects of dietary TSA, a pharmaceutical HDAC inhibitor, were similar. These observations support our hypothesis that SFN acts as a chemical defense by disrupting the epigenetic control systems of herbivorous insects.

### 2. *T. ni* is more resistant to SFN

Interestingly, the specialist herbivore *T. ni* was at least partly resistant to the effects of SFN. The development of *T. ni* larvae was rarely affected by SFN, whether the substance was

consumed in the diet or applied topically to eggs. Similarly, consumption of SFN did not inhibit HDAC enzyme activity in this species. In fact, HDAC enzyme activities remained high in fat body tissues despite consumption of SFN and/or the known HDAC inhibitor TSA.

The difference in responsiveness between *S. exigua* and *T. ni* is not surprising. Herbivores often possess different co-evolutionary adaptations allowing them to feed on toxic host plants with various levels of success [80,83,89,90,114–116]. We considered how some of these adaptations may explain our results. First, some grazers possess mechanisms to deactivate or redirect the "mustard oil bomb" reactions, preventing the accumulation of products such as SFN. These adaptations would be irrelevant in our experiments because larvae consumed SFN directly. Second, we considered that some specialized grazers possess mechanisms to deactivate isothiocyanates via conjugation in the gut. These mechanisms could have protected larvae from SFN in our feeding experiments, however this would not explain the similar results obtained when SFN was applied topically to eggs (Fig 2B). Similarly, it would not explain why *T. ni* is generally resistant to TSA (Fig 3A). Third, we considered the possibility that the resistance of *T. ni* in our experiments, compared to *S. exigua*, was due to differences in the insects' HDAC enzymes themselves; however, this seems not to be the case. HDAC enzymes isolated from *T. ni* and *S. exigua* were inhibited equally by SFN *in vitro* (Fig 3B). Finally, we considered that *T. ni* consuming foliage containing SFN may compensate by overproducing HDAC enzymes; however, we did not find evidence for this either. The expression of HDAC genes was not increased in larvae consuming SFN or TSA. The lone exception was a gene for a poorly understood HDAC enzyme (HDAC11; S1 Table) which was indeed overexpressed in response to SFN and/or TSA. This by itself seems unlikely to explain our results. It is possible that *T. ni* could increase the level of HDAC enzymes via some post-transcriptional mechanism.

### 3. Sulforaphane induces large-scale changes in gene expression

The consumption of SFN affected the expression of thousands of genes. Overall, the patterns of differentially expressed genes were similar in both species in response to SFN and TSA, suggesting that SFN does indeed act primarily as an HDAC inhibitor when consumed in the diet at natural concentrations. Although the patterns of gene expression were similar, the magnitude of changes was generally greater in *S. exigua* as compared to *T. ni*. (Fig 5). In short, similar genes were affected but to a lesser degree in *T.ni*, corresponding with the resistance of this species that we observed in these experiments.

KEGG pathway analyses indicated that the consumption of SFN and TSA impacted core metabolic processes. Both HDAC inhibitors downregulated many genes associated with energy conversion pathways, including glycolysis, gluconeogenesis, pyruvate metabolism, and oxidative phosphorylation, in both species. HDAC inhibitors are well-known to impact energy conversion pathways in cancer cells [59,117] where they usually downregulate genes associated with glycolysis, thereby counteracting the Warburg Effect [118]. However, our results differ in that genes from virtually all energy conversion pathways were downregulated. We also observed a dramatic upregulation of genes associated with protein synthesis in both species, after consumption of both HDAC inhibitors. A similar result was observed in mouse embryos treated with TSA [119,120]. Finally, we observed the upregulation of cuticular protein genes, in both species, which could have implications for development.

In some previous studies HDAC inhibitors, including TSA, have been observed to lower the expression of HDAC genes [121]. In this sense, these substances were doubly inhibitory: disrupting the activity of HDAC enzymes and downregulating the expression of HDAC genes. However, we did not observe this for most of the HDAC genes identified in *S. exigua* or *T. ni*.

## 4. Extending this work to test hypotheses in the field

These observations demonstrate that dietary SFN can inhibit the HDAC enzymes of herbivores, disrupting gene expression and development. Specifically, we demonstrated that SFN slowed the development of *S. exigua*, an important agricultural pest. In these laboratory experiments, we examined the effects of SFN alone, rather than in combination with other defenses present in cruciferous plants. The next step is to evaluate the effectiveness of SFN as an epigenetic weapon in nature.

Based on these results, we envision three possible ways in which SFN could act as an effective defense against susceptible herbivores in nature. First, SFN may cause larvae to be smaller, for a longer time. According to the *slow growth–high mortality* hypothesis, this would extend the window of vulnerability during which caterpillars are most exposed to predators [122,123]. Indeed, this effect has been observed for caterpillars feeding on cruciferous plants in nature [123]. If this is the case SFN would be most effective as a defense in a natural setting when predators are present. Second, SFN could slow development sufficiently to reduce the number of herbivore generations over the course of a growing season. For *S. exigua* we typically observe delays of 2–3 days from first instar larvae to the emergence of moths. However, in longer experiments examining a complete life cycle, we have observed delays of 5–7 days after consumption of SFN. For comparison, the life cycle of *S. exigua* in the field is approximately 22–28 days, rapid enough to generate six generations during the summer in Florida (USA) [124]. Based on our results, SFN could slow development to the point of eliminating one generation of *S. exigua* caterpillars per growing season.

Finally, SFN could sabotage the phenotypic plasticity–the adaptability–of these herbivores, which may be especially detrimental for insects since they rely upon their remarkable phenotypic flexibility for their ecological success [38,92,125–128]. For instance, herbivores commonly adjust their phenotypes to circumvent the induced defense responses of plants in real-time [1–5,129]. If SFN interferes with phenotypic plasticity, it may be most damaging in combination with other plant defenses. In cruciferous plants, synergistic effects of multiple glucosinolates are well-known. It is also interesting to note that such synergistic combinations of HDAC inhibitors and other chemical agents have been documented in medical studies [130,131].

The effects of SFN on *S. exigua* were most apparent at higher temperatures, suggesting that SFN may be most effective in warmer regions or late in the growing season. This result also indicates that continued climate warming may increase the effectiveness of SFN as an epigenetic weapon against herbivores.

## Supporting information

**S1 Table. Expression of HDAC genes in *S. exigua* and *T. ni*.**
(DOCX)

**S1 Fig. Consumption of SFN altered the expression of genes in fat body tissues of *S. exigua* and, to a lesser degree, *T. ni*.** In a common experiment we cataloged gene expression changes in both species in response to SFN, compared to ethanol alone (negative control) and the pharmaceutical HDACi TSA (positive control). Exposure to dietary SFN reduced the size of *S. exigua* larvae but not *T. ni* larvae, as observed in previous experiments. A two-factor ANOVA confirmed that larval size was impacted by the consumption of SFN and/or TSA (P<0.001) and by species (P<0.001). Letters indicate pair wise differences detected by Student-Newman-Keuls multiple comparisons tests.
(DOCX)

**S2 Fig. Significantly differentially expressed genes in *S. exigua* and *T. ni*.** Color intensity reflects the degree of expression changes, highlighting the greater magnitude of expression changes in *S. exigua*, compared to *T. ni*. Hierarchical clustering based on Euclidean distance of 1,792 *S. exigua* (**A**) and 2,454 *T.ni* (**B**) genes significantly differentially expressed after exposure to either SFN or TSA. Values represent the $\log_2$ read counts per million mapped reads (CPM). Data are scaled such that each gene has mean = 0 and standard deviation = 1.
(DOCX)

## Acknowledgments

We are grateful for the advice and assistance of Amy Witter (Department of Chemistry, Dickinson College), Chad Finkenbinder (Benzon Industries), and undergraduate researchers Whitney Finney, Ashely Groff, Marisa Arreola, Peter Gibson, Kyle Ngo, Caroline Nilsen, and Eryn Nelson.

## Author Contributions

**Conceptualization:** Thomas M. Arnold.

**Formal analysis:** Dana J. Somers, David B. Kushner.

**Funding acquisition:** Dana J. Somers, David B. Kushner, Thomas M. Arnold.

**Investigation:** Dana J. Somers, David B. Kushner, Alexandria R. McKinnis, Dzejlana Mehmedovic, Rachel S. Flame, Thomas M. Arnold.

**Methodology:** Dana J. Somers, David B. Kushner, Alexandria R. McKinnis, Dzejlana Mehmedovic, Rachel S. Flame, Thomas M. Arnold.

**Project administration:** Thomas M. Arnold.

**Resources:** David B. Kushner, Thomas M. Arnold.

**Software:** Dana J. Somers.

**Supervision:** Dana J. Somers, David B. Kushner, Thomas M. Arnold.

**Validation:** Dana J. Somers, David B. Kushner, Thomas M. Arnold.

**Visualization:** Dana J. Somers, David B. Kushner, Thomas M. Arnold.

**Writing – original draft:** Dana J. Somers, David B. Kushner, Alexandria R. McKinnis, Dzejlana Mehmedovic, Rachel S. Flame, Thomas M. Arnold.

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
