## [Decision Letter · Decision Letter 0]

3 Sep 2023

PONE-D-23-22782EPIGENETIC WEAPONS IN PLANT-HERBIVORE INTERACTIONS: Sulforaphane disrupts lepidopteran histone deacetylases, gene expression, and larval developmentPLOS ONE

Dear Dr. Arnold,

Thank you for submitting your manuscript to PLOS ONE. After careful consideration, we feel that it has merit but does not fully meet PLOS ONE’s publication criteria as it currently stands. Therefore, we invite you to submit a revised version of the manuscript that addresses the points raised during the review process. Although both Reviewers found the paper to be very well-written, they offer suggestions/comments on a number of points that could be further clarified or more fully developed (e.g. rationale for adding sulforaphane to egg masses, Figure 1, Table 1, and Figure 5 conclusions). They also suggest providing additional statistical support for some of the data. 

We look forward to receiving your revised manuscript.

Kind regards,

J Joe Hull, Ph.D.

Academic Editor

PLOS ONE

“We are grateful for the advice and assistance of Amy Witter (Department of Chemistry, Dickinson College), Chad Finkenbinder (Benzon Industries), and undergraduate researchers Whitney Finney, Ashely Groff, Marisa Arreola, Peter Gibson, Kyle Ngo, Caroline Nilsen, and Eryn Nelson. Support was provided by Dickinson College’s Research and Development Committee and by National Science Foundation award IOS-2151434.”

“This work was supported by grants from the National Science Foundation (award 2151434 to TA, DS, DK) and the Dickinson College Research and Development Committee (to TA).  The funders had no role in study design, data collection and analysis, decision to publish, or preparation of the manuscript.”

4. We note that Figures 1 in your submission contain copyrighted images. All PLOS content is published under the Creative Commons Attribution License (CC BY 4.0), which means that the manuscript, images, and Supporting Information files will be freely available online, and any third party is permitted to access, download, copy, distribute, and use these materials in any way, even commercially, with proper attribution. For more information, see our copyright guidelines: http://journals.plos.org/plosone/s/licenses-and-copyright.

a. You may seek permission from the original copyright holder of Figures 1 to publish the content specifically under the CC BY 4.0 license.

Reviewers' comments:

Reviewer's Responses to Questions

**Comments to the Author**

1. Is the manuscript technically sound, and do the data support the conclusions?

Reviewer #1: Partly

Reviewer #2: Partly

2. Has the statistical analysis been performed appropriately and rigorously? 

Reviewer #1: No

Reviewer #2: Yes

3. Have the authors made all data underlying the findings in their manuscript fully available?

Reviewer #1: Yes

Reviewer #2: Yes

4. Is the manuscript presented in an intelligible fashion and written in standard English?

Reviewer #1: Yes

Reviewer #2: Yes

5. Review Comments to the Author

Reviewer #1: The manuscript is exquisitely written, with the authors having successfully elucidated the nuances in feeding behavior between specialist and generalist pests in a logical and cohesive manner. However, there are several comments that I would like to offer which, I hope, will assist in further refining the manuscript.

1. The title of the paper, "Epigenetic Weapons in Plant-Herbivore Interactions," implies that the study focuses on the effects of sulforaphane (SFN) on feeding behavior in a specialist Trichoplusia ni and a generalist Spodoptera exigua. However, the results indicate that SFN does not have any negative impact on feeding, development, or mortality in T. ni. It is worth noting that a specialist pest typically causes more damage to a plant than a generalist. As such, the results presented in the paper are insufficient to justify the title. To support the hypothesis that S. exigua consumes more plant tissue than T. ni, a feeding/non-feeding choice assay can be conducted.

2. Similarly, the title "Sulforaphane Disrupts Lepidopteran Histone Deacetylases" is somewhat misleading, as the activities of HDAC enzymes were not inhibited, and development was not disrupted in the lepidopteran insect T. ni. Given that generalists are typically more sensitive to plant toxins than specialists, this finding is noteworthy.

3. It should be noted that the family Cruciferae is now known as Brassicacea and should be updated in the manuscript. The abstract states that cruciferous plants produce sulforaphane (SFN), an inhibitor of nuclear histone deacetylases (HDACs). However, it is important to clarify that SFN is not produced constitutively by Brassicacea plants. It is a product of the Mustard Oil Bomb system of Brassicacea, which is activated by herbivore or mechanical damage.

4. The introduction of the manuscript discusses epigenetic regulation and HDAC inhibitors but fails to fully elucidate the role of SFN in herbivory and the reasoning behind its selection.

5. The illustration in Figure 1, which depicts the basic design of single generation experiments, is somewhat unclear. It could be made more informative and illustrative.

6. To confirm whether SFN affects HDAC, quantitative real-time analysis of HDAC-related genes should be conducted, using appropriate housekeeping genes prior to enzyme activity and RNA seq. This will help to validate the upregulation or downregulation of HDAC genes.

7. SFN feeding slowed the development of S. exigua. This slow larval development can contribute to increased feeding time by the herbivore when feeding on Brassicacea plants. However, pupation and mortality of S. exigua were unaffected by SFN treatments. As such, the results presented in the manuscript are insufficient to justify the title. The title could be revised to reflect the differences in feeding behavior between a generalist and a specialist.

8. It is unclear what the rationale behind SFN application to egg masses is and how this correlates with natural herbivore feeding.

9. The SFN treated and untreated larval development should be compared using a paired t-test that student's t-test.

10. Interestingly, the emergence of T. ni moths was not delayed but rather was accelerated by approximately two days in both TSA and the highest concentrations of SFN (Figure 2D). Despite these effects on larval development, there were no detected differences in pupal mass or the size, appearance, and behavior of adult moths in any treatment group. The authors have not provided an explanation for this observation.

11. In Figure 3B, nuclear and cytoplasmic HDAC enzymes were extracted from S. exigua (red) and T. ni (blue), and their activities were quantified in vitro in the presence of various concentrations of SFN. The artificial diet containing the dissolved ethanol concentration is not shown, but its inclusion will help to elucidate the exact effect of SFN on nuclear and cytoplasmic HDAC enzymes.

12. Table 1 can be presented more informatively in the form of graphs rather than a table.

13. Third, we considered the possibility that the resistance of T. ni in our experiments compared to S. exigua was due to differences in the insects’ HDAC enzymes themselves; however, this does not seem to be the case.

Reviewer #2: This manuscript presented the ability of defensive plant products in altering global gene expression patterns of different insect herbivores through inhibiting the activity of nuclear histone deacetylase (HDAC) enzymes. The findings were interesting and would provide new insights into the insect-plant interaction.

Major points:

1. For the INTRODUCTION, I would suggest to provide necessary background first and propose the hypothesis and reasoning at the end. Too much reasoning was mixed with the background information, which might influence the flow of the INTRODUCTION. In addition, I do not think DNA methylation and non-coding RNA is necessary to appear. However, the detailed information for the histone modifications would be required.

2. The authors just used the fat body to quantify HDAC enzyme activities or perform RNA-seq. Because midgut is also an important tissue in plant-insect interaction, I would suggest to provide some reasoning and discussion for such practice.

3. The authors claimed that the magnitude of the response to the HDAC inhibitors was greater in Spodoptera exigua compared to Trichoplusia ni (Figure 5A). Can the authors provide some more details on how to get this conclusion? In addition, the authors identified 181 and 409 genes in S. exigua and T. ni, respectively. What did this mean? Since SFN inhibited the HDAC enzymes in S. exigua but not T. ni, how could this type of comparison shown in Venn diagram make sense?

Minor issues:

L136, please specify which species was studied for the 18 recognized HDAC enzymes.

L220, please clarify in what cases ovaries were also collected?

L320, α=0.05?

L341, the appearance of section “effects of SFN on both species under different temperature” should be earlier than “HDAC enzyme activity detection”.

6. PLOS authors have the option to publish the peer review history of their article (what does this mean?). If published, this will include your full peer review and any attached files.

Reviewer #1: No

Reviewer #2: No

---

## [Author Response · Author response to Decision Letter 0]

21 Sep 2023

Thank you for the helpful reviews and comments. We provided a detailed response to each question and comment in the file "Response to reviewers" uploaded with the revised manuscript. We have uploaded a revised manuscript, with and without track changes, as requested.

---

## [Decision Letter · Decision Letter 1]

4 Oct 2023

EPIGENETIC WEAPONS IN PLANT-HERBIVORE INTERACTIONS: Sulforaphane disrupts histone deacetylases, gene expression, and larval development in Spodoptera exigua while the specialist feeder Trichoplusia ni is largely resistant to these effects.

PONE-D-23-22782R1

Dear Dr. Arnold,

We’re pleased to inform you that your manuscript has been judged scientifically suitable for publication and will be formally accepted for publication once it meets all outstanding technical requirements.

Kind regards,

J Joe Hull, Ph.D.

Academic Editor

PLOS ONE

Additional Editor Comments (optional):

Reviewers' comments:

Reviewer's Responses to Questions

**Comments to the Author**

1. If the authors have adequately addressed your comments raised in a previous round of review and you feel that this manuscript is now acceptable for publication, you may indicate that here to bypass the “Comments to the Author” section, enter your conflict of interest statement in the “Confidential to Editor” section, and submit your "Accept" recommendation.

Reviewer #1: All comments have been addressed

Reviewer #2: All comments have been addressed

2. Is the manuscript technically sound, and do the data support the conclusions?

Reviewer #1: Yes

Reviewer #2: Yes

3. Has the statistical analysis been performed appropriately and rigorously? 

Reviewer #1: Yes

Reviewer #2: Yes

4. Have the authors made all data underlying the findings in their manuscript fully available?

Reviewer #1: Yes

Reviewer #2: Yes

5. Is the manuscript presented in an intelligible fashion and written in standard English?

Reviewer #1: Yes

Reviewer #2: Yes

6. Review Comments to the Author

Reviewer #1: Author has largely provided satisfactory responses to most of the queries. Manuscript can be accepted.

Reviewer #2: (No Response)

7. PLOS authors have the option to publish the peer review history of their article (what does this mean?). If published, this will include your full peer review and any attached files.

Reviewer #1: No

Reviewer #2: No

---

## [Editor Report · Acceptance letter]

10 Oct 2023

PONE-D-23-22782R1 

EPIGENETIC WEAPONS IN PLANT-HERBIVORE INTERACTIONS: Sulforaphane disrupts histone deacetylases, gene expression, and larval development in Spodoptera exigua while the specialist feeder Trichoplusia ni is largely resistant to these effects. 

Dear Dr. Arnold:

I'm pleased to inform you that your manuscript has been deemed suitable for publication in PLOS ONE. Congratulations! Your manuscript is now with our production department. 

Kind regards, 

on behalf of

Dr. J Joe Hull 

Academic Editor

PLOS ONE